# Unmanned Aircraft System Applications in Damage Detection and Service Life Prediction for Bridges: A Review

**Hongze Li [1,2], Yanli Chen [1], Jia Liu [1,2], Zheng Zhang [1,2] and Hang Zhu [1,2,\*]**

1 School of Mechanical and Aerospace Engineering, Jilin University, Changchun 130022, China
2 Key Laboratory of CNC Equipment Reliability, School of Mechanical and Aerospace Engineering, Ministry of Education, Jilin University, Changchun 130022, China
\* Correspondence: hangzhu@jlu.edu.cn; Tel.: +86-1808-866-5997

**Abstract:** The increasing need for inexpensive, safe, highly efficient, and time-saving damage detection technology, combined with emerging technologies, has made damage detection by unmanned aircraft systems (UAS) an active research area. In the past, numerous sensors have been developed for damage detection, but these sensors have only recently been integrated with UAS. UAS damage detection specifically concerns data collection, path planning, multi-sensor fusion, system integration, damage quantification, and data processing in building a prediction model to predict the remaining service life. This review provides an overview of crucial scientific advances that marked the development of UAS inspection: underlying UAS platforms, peripherals, sensing equipment, data processing approaches, and service life prediction models. Example equipment includes a visual camera, a multispectral sensor, a hyperspectral sensor, a thermal infrared sensor, and light detection and ranging (LiDAR). This review also includes highlights of the remaining scientific challenges and development trends, including the critical need for self-navigated control, autonomic damage detection, and deterioration model building. Finally, we conclude with a brief discussion regarding the pros and cons of this emerging technology, along with a prospect of UAS technology research for damage detection.

**Keywords:** damage detection technology; multi-sensor fusion; damage quantification; autonomic damage detection; deterioration model building

## 1. Introduction

A reliability analysis of both products and systems is an important task, closely linked to security and service life [1]. Service safety, traffic safety, and a long service life are of crucial importance in bridge engineering. Optimization of traffic loads consequently has a positive effect on the service life of a bridge. Existing bridge structures and their components are subject to various loads caused by daily traffic. Bridge fatigue is then accumulated by the variable amplitude normal stress cycles [2]. In addition, corrosion is an influential factor in the deterioration of reinforced concrete (RC) structures [3–5]. As a result, it is necessary to recognize structural conditions, such as cracks and pits, in a timely manner to ensure safety.

Over the past decade, the pursuit of novel methodologies and advanced technologies has expanded far beyond rough damage detection. The key enabling concept is multi-scale, accurate measurement of a damaged structure. This includes both efficient detection methodologies and integrated service life prediction based on fatigue and corrosion. Traditionally, visual inspection is the most common way to assess operation condition and to detect defects on structure surfaces. However, visual inspections can be labor-intensive, subjective, and time-consuming, even for well-trained inspectors.

To solve this issue, emerging techniques including structural health monitoring (SHM) were designed for long-span bridges in the past [6]. SHM can be applied to examine the

damaged area and to provide information regarding detected damage for maintenance and repair [7]. By using sensing techniques, SHM systems monitor the structural condition of a bridge in real time and analyze structural characteristics according to data output [8]. An SHM system used to evaluate structural health typically consists of hundreds or thousands of sensors and peripheral processors [9–11]. SHM systems have increasingly been installed in multi-type bridges worldwide for almost 40 years with the aim of acquiring damage data for a bridge [12–14].

These damage data were extensive and hard to extract effective information from when evaluating the condition of a bridge. For the development of emerging light-weight, small, high-performance sensors, an unmanned aerial vehicle (UAV) equipped with sensor(s) is making enormous advances in solving problems related to the conventional procedures. A UAV equipped with special sensors, such as visual cameras and LiDAR, will be more reliable and efficient, as well as simple to operate. An interesting study was conducted which employed a UAV to acquire data for SH. A full analysis of this new method was undertaken and it was compared to conventional inspection methods [15]. Mader et al. [16] developed and constructed a fleet of UAVs with different sensors for three-dimensional (3D) object data acquisition. This contribution highlighted the potential of UAVs in building inspections.

Service life prediction of bridges is a key subject in civil engineering. The damage of bridges stems from two major contributors. Firstly, the service life of bridges is affected by the daily growth of traffic [17]. Secondly, corrosion is an essential factor which accounts for the deterioration of RC structures on a bridge, leading to a corrosion pit in the bridge [3–5]. The coupled effect of fatigue and corrosion is very complex, and refers to chloride ingress, concrete crack expansion, corrosion pit growth, environmental variation, and local stress concentration [18].

There are two approaches commonly employed in bridge structures for service life prediction and fatigue damage evaluation. The first approach is a traditional S-N curve, containing the number of failure cycles (N), and the constant amplitude stress range (S), which conform to an S-N curve by accumulative fatigue experiments. The relationship between S and N is a linear damage hypothesis that extends to variable amplitude loadings, called Miner's rule [19]. Miner's rule tracks fatigue reliability by monitoring data [20–22] or finite element analysis (FEA) calculations [23]. The peak over threshold (POT) approach predicts the load effects of extreme amplitude based on extreme values of monitoring data [17]. The second approach, known as fracture mechanics, explores the crack initiation and growth stage under a stress concentration field [24]. In bridge fatigue calculation and service life prediction, two approaches can be applied sequentially. Firstly, the S-N curve method is employed to determine fatigue life. Secondly, a fracture mechanics approach is utilized for refining a crack-based remaining service life assessment and maintenance strategies.

For image-based crack detection [25–29], edge detector-based approaches (e.g., Roberts, fast Haar, Fourier, Sobel, Butterworth, Canny, Prewitt, Laplacian of Gaussian, and Gaussian) [30–36] and deep learning-based approaches [37–40] are the two most widely implemented approaches. Edge detectors have been proposed to detect cracks, and their performance depends on the thresholding method and the image gradient formulation. The performances of the latter two approaches in detecting cracks have been evaluated and compared [27,41]. Another approach suggested using 3D LiDAR [27,42–45] as a remote sensing technique for damage detection by 3D construction. In addition, the multispectral and hyperspectral sensors have also been applied; they cover a wider range of wavelengths compared to visual or thermal cameras [46,47]. Multi-sensors and data fusion increase functionality and accuracy by combining several types of information [46,48,49].

Algorithmically, how multi-sensor data can be mapped to damage detection remains unclear, even though the potential of UAV multi-sensor data fusion has been proven. Meanwhile, how to design them to facilitate high-resolution, adaptive systems, alongside their integration and transition, remains unclear. This review provides a roadmap detailing work on damage detection, considerations, future areas of research, and opportunities

for crack growth. The rest of this paper is organized as follows: in Section 2, underlying platforms and peripherals applied to gather damage information are listed in detail; in Section 3, data processing methods commonly used to handle or extract valid information is introduced; in Section 4, several bridge service life prediction models, including crack extension, fatigue failure, and material corrosion are reviewed; in Section 5, we provide some considerations when utilizing this emerging technology; in Section 6, we discuss advantages, limitations, and future needs of UAS bridge damage detection; and in Section 7, the conclusion and opportunities are given. The methodology and structure of the articles covered in the main elements of this review are shown in Figure 1.

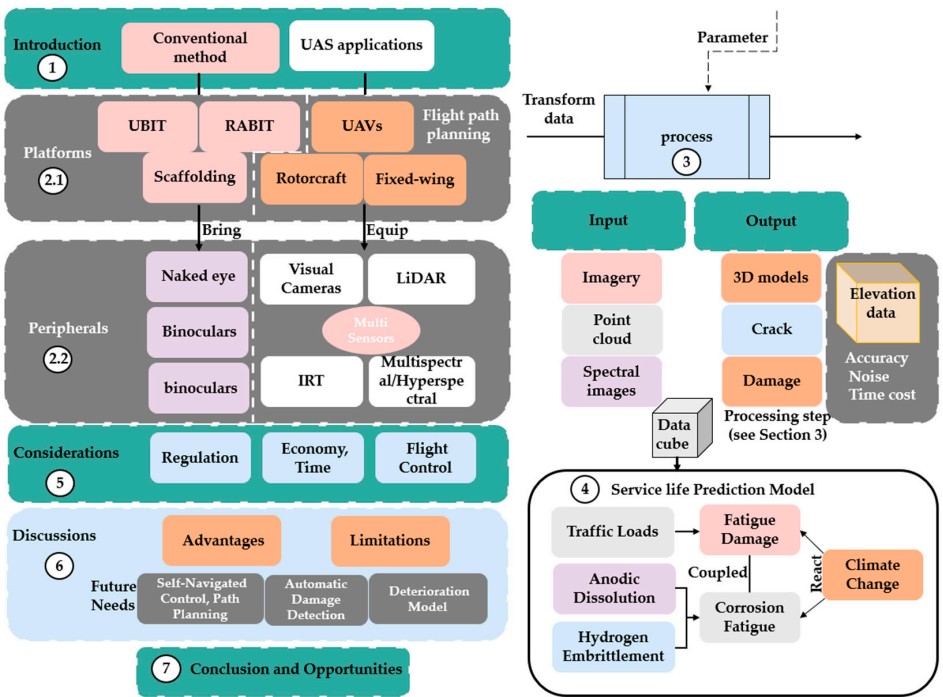

**Figure 1.** Flow chart of the methodology and structure of the article depicting the main elements of the study.

## 2. Platforms and Peripherals

UAS damage detection needs physical components, including the underlying platforms (UAV) and a few common peripherals (visual cameras, LiDAR, thermal infrared sensors, sound navigation and ranging, magnetic sensors, multi-sensors, and data fusion).

### 2.1. Platforms

The term "platform", in relation to the underlying UAV body, facilitates UAS inspection by equipping external sensors. As these UAVs began to incorporate more peripheral technologies and grew in complexity, a new term was developed to describe the whole system together—UAS. UAS was defined as a combination of UAV, either rotorcraft or fixed-wing aircrafts, the ground control system, and the payload (what it is carrying). When it comes to bridge inspection, rotorcrafts and fixed-wing aircrafts are all available platforms to choose.

UAS inspection has clear advantages compared with traditional inspection methods: higher security level, higher information completeness, more efficiency, lower cost, less labor, and more types of information. In some special areas hard to reach by manpower, UAS allows easier access, and path preplanning-based computers can provide improved site visibility and optimized views [33,50]. A financial advantage is of great significance when building a UAS equipped with both LiDAR [51] and image sensors [52]. In addition, various single-sensor technologies or multiple sensors in combination can be utilized for different inspection purposes.

Although there are many available platforms, such as kites and balloons, UAVs are much more variable in their technical specifications. The requirements for lifting capability and platform safety are less stringent with UAVs; the cost and technical expertise needed to operate such systems also vary greatly. When compared with conventional platforms, another advantage of UAVs is that they can be used in situations that require delicate control through specially designed dynamic systems. This level of control is always necessary for narrow space detection in bridge inspection missions. Technical obstacles will not bother users; there is a vast body of useful internet resources (e.g., diydrones.com, dronezon.com, rcgroups.com, accessed on 21 August 2022) as well as practical guidebooks [53] with up-to-date information on UAS technology, models, operational use, navigation tutorials, applications, and economic potential. The peer-reviewed academic literature may also help, containing UAS special issues and proceedings of dedicated UAV conferences (e.g., Remote sensing [54], GIScience and Remote Sensing [55], and International Journal of Remote Sensing [56]).

Traditional visual inspectors usually perform the task by walking on the deck, using binoculars or relying on their naked eye to judge the damaged area. In regions that are difficult to access, scaffolding or big mechanical equipment such as under-bridge inspection trucks (UBIT) may be useful [57]. In 2014, the robotics-assisted bridge inspection tool (RABIT) was the first robotic vehicle used for surface detection and subsurface detection on a bridge deck [58,59]. In the past, fixed-wing UAVs were needed to mount heavy equipment, such as a 3D LiDAR, depending on mounted sensors [60]. However, with the development of autonomous driving, the 3D LiDAR has been updated to a smaller size and lighter weight to enable the equipment of LiDAR and other sensors to be carried on rotorcrafts. Rotorcrafts are most commonly used for their low cost, flexibility, and ability to hover. For bridge inspection, the most common UAV configurations include quadcopters (four propellers) [61–64] and hexacopters (six propellers) [61,62,65]. A fleet of UAVs with different types of sensors has been developed and constructed [16]. The different types, inspection methods, advantages, and shortcomings of different bridge detection platforms are shown in Figure 2 and Table 1.

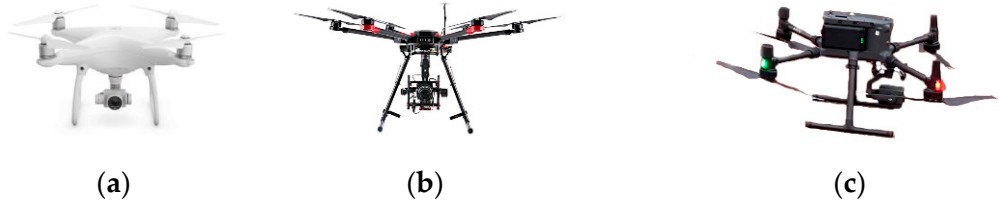

(**a**) (**b**) (**c**)

**Figure 2.** Different types of UAVs used in bridge detection. (**a**) DJI Phantom 4. (**b**) DJI M600 Pro. (**c**) DJI M300 equipped with Zenmuse H20.

**Table 1.** Inspection method, advantages, and shortcomings of bridge detection platform.

| Platforms | Ref. | Focus | Method | Main Advantages | Main Shortcomings |
|---|---|---|---|---|---|
| UBIT | [57] | Under bridge, bridge piers | Manual recording | Easy to implement, detect inaccessible areas | Expensive, threatened inspector's security |
| RABIT | [58,59] | Bridge surface | Ultrasonic echo, vision | Labor saving, picture record | Deck inspections only |
| Fixed-wing UAV | [60] | Overall bridge | External sensor (s) | High efficiency, low cost, and labor saving | Hover-limiting |
| Rotorcrafts | [57,62–65] | Overall bridge | External sensor (s) | Hovering, no runway is required | Sensor weight restriction |
| UAVs fleet | [15] | Overall bridge | Multi-sensors | Various types of sensors | Hard to cooperate with different UAVs |

UBIT: under-bridge inspection truck, RABIT: robotics-assisted bridge inspection tool, UAV: unmanned aerial vehicle.

The above-mentioned platforms are usually controlled by computer programming in combination with a global navigation satellite system (GNSS), such as global positioning systems (GPS), to acquire data, no matter what type of sensor is equipped. Many professional-grade UAS are now equipped with real-time kinematic (RTK) or post-processing kinematic (PPK) GNSS, providing the high accuracy needed for flight control, together with attitude information from the inertial measurement units (IMU). The UAV may be flown manually by experienced pilots, semiautomated by the less experienced, or in fully autonomous mode, from hand-launching to automatic landing. In fully autonomous mode, flight path planning can be accomplished with flight-planning software, where the aircraft follows a predefined flightline along a set of waypoints. Both methods are applied to bright inspection surveys, on the basis of computing exposure interval, flying height, ground sample distance (GSD), flight-path orientation and spacing, and photogrammetric overlap, manually or automatically [66,67].

UAV flight path programming has recently been introduced as an effective method to provide the potential locations of surface defects; it efficiently achieves perpendicular and overlapping views for sampling the viewpoints, with the help of powerful algorithms such as a spanning tree algorithm [68], a wavefront algorithm [69], and a neural network [70]. There are other methods aiming at different purpose, such as the traveling salesman problem (TSP) [71,72] and the A* algorithm [73] for minimizing the chances of the path passing through pre-defined viewpoints; the art gallery problem (AGP) [74] for minimizing the number of viewpoints; and coverage path planning [50,75] for 3D-mapping. The core objectives of flight path programming are improving safety, accuracy, coverage, and efficiency and minimizing flight cost.

### 2.2. Peripherals

#### 2.2.1. Visual Cameras

Visual cameras are commonly used in UAS inspections, with inspectors on the ground to operate the UAV and onboard imaging sensors and cameras [76]. Computer vision techniques are utilized in automated visual inspection quantification systems for bridge damage detection introduced by many researchers [77–82]. Professional remote sensing UAVs, such as the DJI phantom and Mavic series [57,66] (DJI innovation, Shenzhen, China), can be directly used to capture damage information from the surface of a bridge. Digital cameras, such as Nikons, were selected as visual sensors to acquire high-resolution images [62]. Three general image-based inspection methods are raw image direct inspection, image enhancement technology, and image processing algorithms. Compared to image enhancement technology and image processing algorithms, manual raw image direct inspection is prone to inaccuracy and is time consuming [41].

#### 2.2.2. LiDAR Sensor

The terrestrial 3D LiDAR scanner is a piece of optical sensing equipment specially used for distance measurement, and it has been widely used in remote sensing. Millions of position data points collected by LiDAR can populate a surficial area. These position data points can be used to reconstruct 3D models and object mapping of the interest area [43,83]. The feasibility and advantages of merged UAV–LiDAR systems have been demonstrated [84,85].

#### 2.2.3. Thermal Infrared Imagery

Infrared thermography (IRT) imaging technology has been widely used in non-destructive evaluations (NDE), examinations of bond defects in composite materials [86], complicated geometries [87], moisture content in roofs, performance analysis of wet insulation [88], post-disaster inspections [89–91], and crack, nugget, expulsion, and porosity in concrete decks [92–94]. Both active [95,96] and passive [85,97] thermography can identify internal defects of bridges, such as delamination in bridge piers and decks. Sakagami [98] developed a new, thermal non-destructive testing (NDT) detection technique for bridge

fatigue crack detection by detecting a temperature change. Based on temperature distribution data, Omar [99] and Wells [100] completed a concrete bridge inspection using a UAS mounted with infrared thermography. Escobar-Wolf et al. [101] used IRT for undersurface delamination and deck hole detection, which generated thermal and visible images for a 968 m$^2$ area. Compared with direct contact hammer sounding, an overall accuracy of approximately 95% showed in 14 m$^2$ delamination. The three types of sensors below, mainly equipped in UAS, are shown in Figure 3 and Table 2.

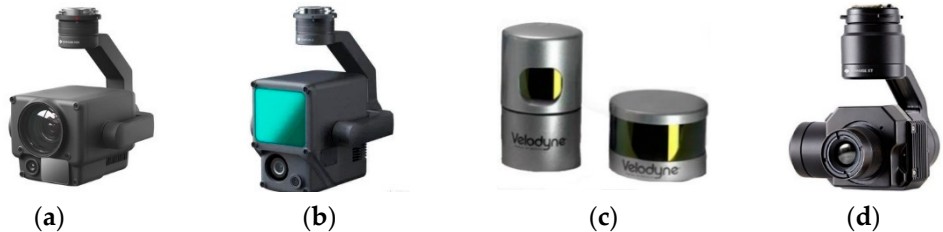

| (a) | (b) | (c) | (d) |

**Figure 3.** Different types of peripherals used for bridge detection. (**a**) DJI Zenmuse H20. (**b**) DJI Zenmuse L1. (**c**) HDL-32E, VLP-16 LiDAR. (**d**) DJI Zenmuse XT.

### 2.2.4. Multispectral and Hyperspectral Sensors

Recent interests have included the study of overall spectral information acquisition and generation [102–104]. Compared to visual and thermal cameras, multispectral and hyperspectral imaging can obtain a bridge inspection image in several bands because they cover a wider wavelength range. Visual and thermal cameras collect intensity for only three bands: red, green, and blue (RGB) colors. Kim proposed a damage detection algorithm and tested it by decomposing a efflorescence area of two concrete specimens based on their spectral features [105].

### 2.2.5. Multi Sensors

With the emergence of new sensor technologies, how to effectively integrate and combine several sensing technologies capable of collecting and processing detailed information has been a frontier problem. The concept of airborne LiDAR and hyperspectral integrated systems was first introduced by Hakala et al. [106]. They acquired 3D model and image information of an observed area using an integrated multispectral-LiDAR imaging system. The LiDAR and hyperspectral integrated systems based on a UAV was designed to integrate two imaging technologies effectively and collect simultaneous 3D model and spectral information accurately [46]. Mader et al. [16] organized a fleet of UAVs equipped with different sensors (LiDAR, camera, and multispectral scanner) to investigate the potential of UAV-based multi-sensors in a data feasibility analysis.

### 2.2.6. Other Sensors

Alongside the above common sensors, other potential sensors, such as the magnetic sensor, were also applied to UAS inspection for special demands or assignments. Magnetic sensors with sufficient power and accuracy can be applied to generate defect maps of ferrous materials in a bridge. Climbing robots equipped with magnetic wheels have been developed to measure the vertical component of an induced magnetic field [107,108]. The limitation of magnetic sensors is their testing capability, and that the test material only includes steel bridges. For underwater bridge structure inspection, sound navigation and ranging (SONAR) was a common testing method [109,110]. Vertical take-off and landing platforms can be used for close inspection, along with rejecting environmental disturbances [111].

**Table 2.** Work mode, main advantages, and shortcomings of peripherals used for bridge detection.

| Peripherals | Ref. | Focus | Work Mode | Main Advantages | Main Shortcomings |
|---|---|---|---|---|---|
| Visual cameras | [65,76–82] | Visible bridge damage | Image capture | Easy access to data | Appropriate shooting angle and advanced path planning |
| LiDAR | [43,84,85] | Bridge damage structure | Transmitting and receiving laser light | Scanning efficiency, overall point cloud data | Expensive, large data, seriously affected by vibration |
| Thermal infrared imagery | [86–100] | Internal defects | Active and passive thermal imaging | Internal defect identification | Hard to map processing and threshold extraction |
| Multispectral and hyperspectral | [102,103,105] | Spectral information | Push-broom scanning | Wider range of wavelengths | Complex dimension and noise reduction |
| Multi-sensors | [15,46,108] | Various data types | Multi-sensors cooperate | Various types of sensors | Hard to achieve data fusion |

LiDAR: light detection and ranging.

## 3. Data Processing

UAS are able to collect various types of data using external sensors, and the next step is usually extracting useful information through proper data processing methods. This section is a generalization of the data processing approach after data has been collected by platforms (multi-type UAVs) and various sensors (visual cameras, LiDAR, and so on).

### 3.1. Three-Dimensional Reconstruction

Researchers are able to construct 3D models of bridges and identify delamination in the deck based on UAS data. These 3D reconstruction types can be divided into three approaches according to their respective data resources: image based [44,79,112–114], LiDAR based [115–117], and image and LiDAR merged [118,119]. Photogrammetric point clouds can be acquired by multi-view images [120,121]. Oude Elberink [122] proposed an automated 3D reconstruction aimed at highway interchanges by data integration of point cloud and topographic maps. Recently, deep learning was used to reconstruct 3D models of cable-stayed bridges by Hu et al. [123]. This approach identifies bridges in panoramic images, then decomposes bridges into parts by utilizing deep neural networks. The proposed method has a higher spatial accuracy in comparison to the manual approach. In addition, the method is robust to noise and partial scans.

After an accurate model is established, damage detection can be solved by auto-clustering algorithms, such as k-means or density-based spatial clustering of applications with noise (DBSCAN). However, missing data is the most common source of error in 3D-reconstructed point clouds, caused by site-line-based occlusions [124], nonuniform data densities [125], inaccurate geometric positioning [126], surface deviations [127], and outlier-based noise [128]. To solve these problems, 3D reconstruction optimization algorithms and noise removal techniques should be developed to improve the performances beyond commercial software in the construction of complex objects. Another issue in 3D reconstruction by images is the high time requirement stemming from both data collection and model building. Chen et al. [44] required approximately five hours for total model building and extra images to increase the time past the minimum. However, previous testing indicated that 48 images was enough to reconstruct a detailed bridge model [129]. Although the inspected object was small, $140 \times 53 \times 23$ (H $\times$ W $\times$ D) cm, one hour was needed to create the model [130]. Free or commercially available 3D software (Automatic Reconstruction Conduit, Agisoft PhotoScan, and Pix4D [131,132]) can be used to construct 3D models. For damage detection using reconstruction models, accuracy usually depends on model precision and a negative correlation with UAV flight height. In a recent damage evaluation study, the 3D volume calculation showed a 3.97% error from the UAV 10 m data set, and a 25% error from the UAV 20 m data set [44].

*3.2. Image-Processing Techniques*

Aimed at different goals or different imaging patterns, the image processing algorithms were also different. For field image generation, various distortions, interferences, and vignette corrections are essential based on raw format images with geo-referencing [133]. Image processing algorithms can be utilized in crack identification based on concrete bridge images. Many researchers proposed crack assessment diagram frameworks to realize crack quantification and localization. The proposed diagram framework always follows a combination of several image processing techniques, and the common process can be categorized as follows. Firstly, a pretreatment of images containing a region of interest (ROI) is acquired. Following this, the ROI image is enhanced with the noise removed. Finally, cracks are identified by an edge filter (Fujita [134], Sobel [34,135], Laplacian of Gaussian [32], Butterworth, Canny [136], and Gaussian [137–139]), and are subsequently labeled or saved.

Xu [140] developed a crack detection method utilizing the Boltzmann machine algorithm based on bridge images. Ahmed et al. [34] detected major cracks using the Otsu approach after utilizing Sobel's filtering to eliminate residual noise. An edge filter in the image processing algorithm classified deck cracks as edges for two-dimensional (2D) images and then extracted position information to calculate crack length and area. Clustering algorithms, such as c-means [141–143], k-means [144,145], and DBSCAN [146] are also common for concrete crack detection. In short, choosing from the large number of methods and algorithms for crack detection creates confusion. As a result, comparisons between different image processing techniques through the same samples have been carried out by many researchers. Abdel performed a comparative study using four edge detection methods—fast Fourier transform (FFT), Canny, Sobel, and fast Haar transform (FHT) based on 25 images of defected concrete [27]. Cha [40] compared Sobel, Canny, and a convolutional neural network with four images. In addition to images, LiDAR point cloud data can be projected onto a 2D plane for convenience [31]. The image processing algorithms mentioned above accurately detected 53–79% of cracked pixels, but they produced residual noise in the final binary images.

*3.3. Deep Learning in Neural Networks*

Currently, deep learning methods have risen in popularity in order to make sense of damage datasets collected by UAS. A deep learning architecture is composed of various modules; they compute non-linear, input-output mappings and are subject to learning. The backpropagation procedure to compute the gradient of an objective function with respect to the weights of a multilayer stack of modules is nothing more than a practical application of the chain rule for derivatives (Figure 4) [147].

Deep convolutional neural networks (DCNNs) and convolutional neural networks (CNNs) have been increasingly utilized in bridge damage detection for image processing, and different architectures are increasingly used in the automatic identification of structural defects in images [39,148,149]. Compared to most traditional methods, damage detection by CNNs avoids the disadvantages of requiring hand-crafted features. Traditional methods can only identify certain defect types and the results are susceptible to surrounding disturbances [150]. An early study concerning this can be obviously seen in the research by Munawar, which contained a vast dataset of 600 images using pixel-wise segmentation prediction and crack detection in buildings in Sydney [151]. Chen et al. [152] proposed an NB-CNN model to detect cracks in video frame, where a 98.3% hit rate was reached by gathering information from video frames. Hoskere et al. [153] recognized six different types of damage using a damage localization and classification technique modified by two classic CNN architectures, ResNet and VGG19. Cha et al. [40] designed a deep architecture of CNNs and recorded that accuracies were 97.95% in 8 K images and 98.2% in 32 K images after training and validation. Significant improvements regarding test speed were discussed in a separate study [42].

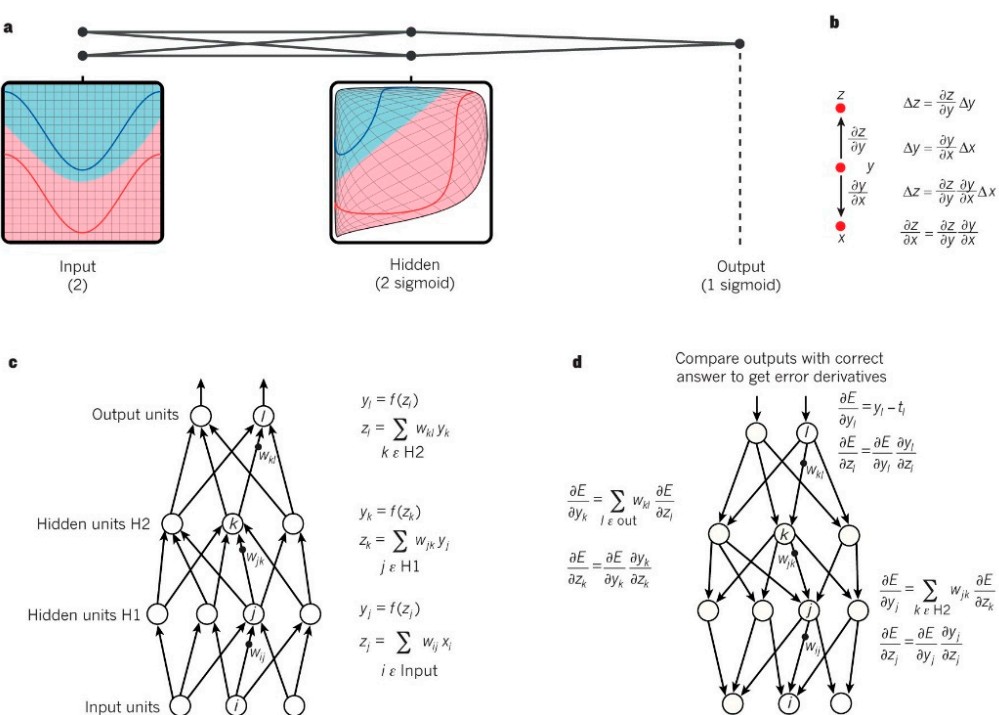

**Figure 4.** Multilayer neural networks and backpropagation showing (**a**) an illustrative example with only two input units, two hidden units, and one output unit in a multi-layer neural network; (**b**) the chain rule of derivatives, showing two small composed effects including $\Delta x$ in $x$ gets transformed first into a small change $\Delta y$ in $y$ by getting multiplied by $\partial y/\partial x$ and $\Delta y$ in $y$ creates a change $\Delta z$ in $z$, similarly; (**c**) the equations used for computing the forward pass in a neural network with two hidden layers and one output layer,; and (**d**) the equations used for computing the backward pass.

Gong et al. [154] identified collapsed buildings by three machine learning classifiers: the K-nearest neighbor (KNN) method, random forest, and support vector machine (SVM). Sun et al. reviewed applications of artificial intelligence techniques (AI) in bridge damage detection and discussed the challenges, upper limit, and future trends [6]. The performance of DCNN and six common edge detectors (Roberts, Sobel, Prewitt, Laplacian of Gaussian, Gaussian, and Butterworth) for crack detection of bridge concrete images was compared in a separate report [82]. In regards to identification precision accuracy, the neural network was capable of detecting concrete cracks wider than 0.08 mm after trained modes and 0.04 mm after transfer learning mode. The required time, main advantages, and shortcomings of different data processing methods can be seen in Figure 5 and Table 3. The above studies show a higher accuracy compared to image processing technologies, usually exceeding 90% after several days of training.

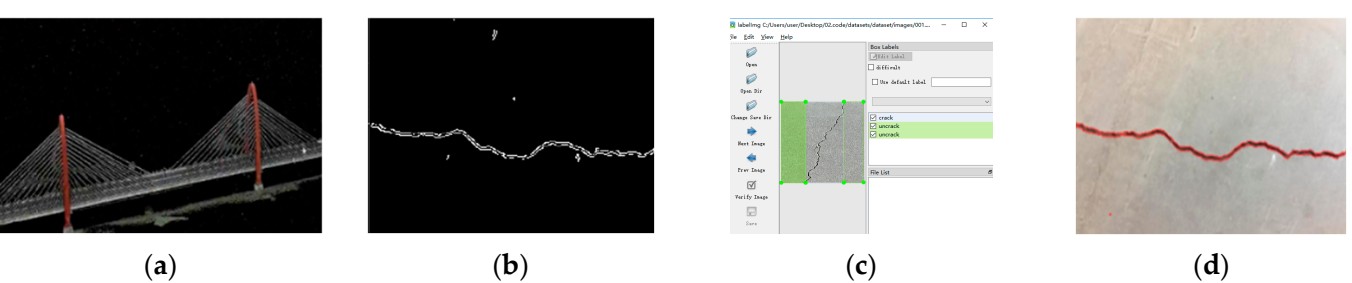

(**a**)    (**b**)    (**c**)    (**d**)

**Figure 5.** Results from different data processing methods, showing (**a**) cable-stayed bridges, (**b**) Canny algorithm, (**c**) labelling crack region for deep learning, and (**d**) threshold extraction and painting.

**Table 3.** Required time, main advantages, and shortcomings in different data processing methods.

| Data Processing | Ref. | Focus | Required Time | Main Advantages | Main Shortcomings |
|---|---|---|---|---|---|
| 3D reconstruction | [43,64,98–118] | Imagery and point clouds data | Hour level | Complete bridge detection, detailed model generating | Time consuming, high precision required |
| Image-processing techniques | [88–90] | Imagery | Hour level | Scanning efficiency, overall point cloud data | Complicated, hard to select |
| Deep learning | [92–106] | Image dataset | Day level (model training) | High precision | Training model in advance |
| Threshold extraction | [14,49,96] | Spectral images | Hour level | Internal defect identification | Hard to extract threshold, sensitive to temperature |

## 4. Service Life Prediction Model

After damage data processing, the next step is to undertake a condition assessment and service life prediction according to existing damage. This is necessary to repair or even destroy the bridge for public security consideration if the bridge displays poor reliability. The operation service life of bridges is affected by variable traffic loads, climate change, and chloride ingress, variables that are intercoupled and time-varying. An accurate service life prediction model, deduced from the deterioration model, makes sense in calculating bridge lift and providing corresponding maintenance strategies. This section presents a literature review of bridge deterioration modeling and deduces a service life prediction model containing research on the structural assessment of concrete bridge decks. The content is based on three common models used for bridge service life prediction.

### 4.1. Traffic Load-Fatigue Damage Model

The growth of traffic loads has become a popular topic and has been increasingly studied in recent years. This issue is critical in bridge deterioration for both fatigue crack growth and fatigue damage accumulation [32]. This type of damage is inevitable and is a common concern to public, management agencies, and bridge owners. When assessing fatigue problems affected by traffic loads, a crucial step involves determining a traffic load model. Modeling the traffic loading for same-direction, two-lane, and multi-lane traffic, and the relationships between adjacent vehicles in both lanes, have been researched [155,156].

In the modeling of traffic loads, characteristic load effects were generally acquired from Monte Carlo [157–160] simulations of traffic in each lane simulated independently [160]. The early development of bridge weigh-in-motion (BWIM) [161–165] provides supporting data for fatigue damage assessment and service life prediction using measures and records from long-term authentic responses. Guo provided a fatigue reliability level method in the case of load models based on traffic counts [166]. Simplified simulations through a multi-lane traffic model [164,167,168] and a car stress model [17] were applied due to the complex nature of traffic loads. A long-term traffic load spectrum contains the complete load information, but directly measuring is difficult due to the restrictions of testing technology. In addition, this approach is limited as it is time consuming and expensive.

Therefore, a long-term load spectrum must be obtained according to a short-term one. Considering various forms and characteristics of loads, selection of proper extrapolation methods is of great significance. A review of extrapolation methods has been published [169]. The load effects of extreme amplitude can be predicted by using the POT approach based on extreme values [17,156,170].

Following traffic load modeling, fatigue life prediction must be considered. The stress–life method reveals a relationship between the number of cycles and the stress range

when visible cracking occurs [171,172]. The Basquin function is widely used in fatigue failure [173,174] and can be expressed mathematically by [175]:

$$NS^m = A \qquad (1)$$

Or

$$\log N = -m \log S + \log A \qquad (2)$$

where $m$ and $A$ are positive empirical material constants. In Equation (2), $m$ and $\log A$ are the slope of the curve and the intercept on the $\log N$ axis, respectively.

The rain flow cycle counting method [176,177] can be used to analyze the statistics of stress cycles. In engineering practices, the Miner's rule damage accumulation linear hypothesis is the most common way to predict the fatigue life. According to the rule, it occurs after the fatigue resistance is fully consumed. The failure of component fatigue can be generalized as [19,178,179]:

$$\sum \frac{n_i}{Ni} = 1 \qquad (3)$$

where $n_i$ is the applied cycles among the stress range ($\Delta\sigma_i$) and $N_i$ is the corresponding fatigue life cycles in the stress range of $\Delta\sigma_i$.

### 4.2. Anodic Dissolution and Hydrogen Embrittlement Corrosion Fatigue Coupled Model

As well as traffic load-fatigue, RC bridge corrosion-fatigue has been regarded as an essential failure paradigm for bridges [180]. The interaction between fatigue and corrosion can be seen as a coupled effect, where fatigue and corrosion join forces to deteriorate the performance of bridge RC structures. In the corrosion-fatigue coupled model, hydrogen embrittlement [181] and anodic dissolution [182] are the two main factors which need to be taken into consideration. The factors above lead to compromised steel strength, pitting corrosion, and weakening of the bonds between concrete and steel bars [183]. The corrosion rate can be accelerated and the load-carrying capacity can be reduced by severe corrosion occurring in the cracking areas of bridge concrete [184,185].

The life of a corrosion pit is predicted by experimental crack growth or linear-elastic fracture mechanics, treated as a tunneling crack [186,187]. Early bridge fatigue failures are mainly found on corroded surfaces and take the form of shallow pits; their size and density are estimated by statistical or empirical corrosion pit growth rate or equivalent crack size [188–190]. A coupled corrosion-fatigue model was performed by Ma [20]. In the study, they considered concrete cover severe cracking, traffic frequency, and the effects of seasonal environment aggressiveness. Detailed crack and corrosion pit growth models were provided and the entire procedure included three stages: (1) corrosion initiation, (2) competition between fatigue crack growth and corrosion pit growth, and (3) determination of structural failure.

### 4.3. Climate Change and Bridge Life Prediction Model

Climate change is another important aspect that must not be ignored when calculating the service life of a bridge. One relevant risk involves an increased deterioration and degradation rate of the concrete structure materials in a bridge [191]. For instance, in a recent study [192], the risk of corrosion and damage to the concrete structure of a bridge were quantitatively assessed in the cities of Darwin and Sydney, Australia, under future climatic conditions. A climate simulation of wind speeds was initiated followed by global models concerning emission scenarios [193]. A total of 31 climate change risks were grouped into seven main categories, and the interconnectedness of these risks were discussed and reviewed [194].

## 5. Considerations

This review aims at displaying the concise, but thorough, key scientific advances of the recent development of UAS application in the realm of bridge damage detection. In

addition to the above technical knowledge, there are some considerations that should be taken into account when using this emerging technology.

### 5.1. Regulation

As both UAVs and UAS are emerging technologies, it is difficult to find laws or regulating bodies in many areas. According to the weight of the UAV or UAS, all countries have a classification scheme except for Japan, China, Rwanda, and Nigeria. They follow a basic risk-based concept—flight conditions are strictly limited by the associated risks (i.e., weight). The relevant risks include malfunction and uninterrupted damage to people or property by of UAVs [195]. In essence, regulations and laws seek to minimize the risks and perceived harms, mainly aim at operational limitations. Most nations have defined horizontal distances from people and property, or so-called no-fly zones, which need to be taken into account. UAS flights are prohibited from flying in the vicinity of people and/or crowds of people. Many countries specifically mention minimum lateral distances these devices must be from people, usually ranging between 30 m and 150 m (e.g., Japan, South Africa, Canada, Italy, and the UK). Hence, UAS bridge inspection could not be applied in the above situations due to restrictions in UAV operation. Another general limitation is the maximum flying height; many countries have a minimum of 90 m and a maximum of 152 m, aiming to separate manned aircrafts and UAVs. Although regular inspections are low altitude aerial operations for high resolution and are unlikely to reach 90 m, the long focal length required to create panoramas or inspection for suspended-cable structures on top of suspension bridges are strongly hampered. In different nations, the determination of camera lens focal length is slightly affected by various maximum height levels, that is, to select shorter focal lengths for higher altitude limits. However, the strategy to respond to large traffic volumes may require lane closures. This is also the reason why the majority of the literature targets abandoned bridges. For a deeper understanding, the reader is referred to UAV regulation literature [196–199].

### 5.2. Economy and Time

Traditional damage detection was undertaken using visual inspections, and UBIT was employed in special regions difficult for manpower to access [200]. UAS inspection ($20,000) was proven to be 66% cheaper compared to traditional inspection methods involving four UBIT and a 25-m man lift ($59,000). A 2017 cost analysis of a large-scale (2400 m long) bridge showed that UAS inspection was cheaper than the traditional inspection methods [103]. A further example (ITD Bridge Key 21105) shows a cost of $391 per hour using UBIT inspection and $200 per hour using UAS inspection. In their study, UAS inspection is calculated as almost half of UBIT inspection for the hourly cost. According to the above research, UAS inspection was able to decrease both the time spent and the budget for bridge inspections under the premise of providing high-quality inspection information comparable to traditional inspection. However, the above statistics are based on essentials in the context of equipment; the specialized equipment and cameras cost a lot compared to a UAS platform itself. For instance, a popular consumer UAV, DJI Mavic 3 Pro, equipped with a high-resolution RGB camera, costs approximately $2000, whereas adding a Zenmuse X7 with a camera lens built specially for damage detection could cost nearly $5000 [201–203]. However, as the technology develops and corporate commercialization grows, the cost of these complete systems is forecast to decrease in the future.

### 5.3. Flight Control

General purpose navigation is commonly undertaken by GPS, micro-electro-mechanical systems (MEMs), inertial navigation sensors (INS), altitude sensors (AS), IMU, accelerometers, and gyroscopes through autopilot computers and external sensors [204]. Due to the restrictions of bridge environments, UAV self-controlled navigation is still a large challenge in the domain of bridge inspection [205]. A GPS signal is limited in the environment under the bridge, risking catastrophic technical difficulty to the UAS and external sensors, while

also bringing a security risk to the pilot, the public, and the inspector [206]. Furthermore, weather conditions are an important factor for UAS flight control.

The distance of UAV to the detection surface and the minimum required number of images should be taken into account before planning a flight path. The distance of an UAV to the detection surface can be determined by the width of the narrowest detectable crack and the required size of the imaging scene. The minimum required number of images can be computed by the central angle between the two adjacent positions of the UAS [33]. Paine (2003) recommended 30 $\pm$ 15% for sidelap and 60 $\pm$ 5% for endlap [207] to optimize aerial photography and image interpretation.

## 6. Discussion

This section discusses the advantages, limitations, and future needs for the application of UAS in bridge damage detection. Advantages and limitations provide a reference for users to consider this emerging technology. In addition to the above content, future needs include self-navigated control, path planning, automatic damage detection, and a precision deterioration model.

### 6.1. Advantages and Limitations

Limitations exist in the application of UAS damage detection, although powerful benefits and potentials also exist. A prominent advantage of UAS damage detection technology is that they are capable of creating accurate 3D models and mapping for damage detection by a single sensor or multiple sensors. However, the photogrammetric and point cloud data processing times are very extensive, requiring a lot of computing time when applying high-resolution sensors. Another limitation is UAV flight times (often approximately 30 min for one flight), and an extra backup battery is necessary for large-scale missions. In addition, it would likewise expend more battery power to equip additional sensors. Nevertheless, UAS inspections utilize a faster inspection speed than traditional methods under the premise of requiring a large upfront cost with respect to high solution equipment. As a result, essential care should be taken in evaluating the economic cost when applying UAS damage detection technology.

### 6.2. Future Needs

6.2.1. Self-Navigated Control and Path Planning

Autonomous control was commonly used with some form of appropriate airborne sensors and autonomous control algorithms. However, the limitations with regards to UAS autonomous control involve the rationality of flight path planning which is directly dependent on the professional skills of the pilot. Self-navigated UAS are essential for achieving autonomous path planning and efficient bridge inspection [208]. The breakthroughs in UAS technology containing accuracy sensors and integrated navigation algorithms, flight path planning, inspector safety, and data collection reliability are the crux of self-navigated UAS damage detection.

6.2.2. Automatic Damage Detection

Recently, automatic damage detection technology and algorithms regarding images, point cloud data, and 3D models have been developed to provide a tool for crack detection and quantification, but more robust sensors and data processing tools are urgently needed in addition to the existing inspection framework. Feature detectors and feature descriptors have the ability to detect bridge surface defects using programs such as SIFT and SURF to generate 3D reconstructed bridge models. For example, spalls, cracks, and surface degradation by 3D model reconstructions of a building require highly skilled technicians and are time consuming when using high-resolution sensors. The realm of deep learning is very active in academic research and applies to bridge damage detection. The deep neural network method is an emerging methodology that catches sensitive features to structural condition variations and performs better than traditional algorithms in classification prob-

lems. Hence, quickly locating damaged areas in bridge structures by deep learning may be easily achieved, although their application in various bridges still needs investigation. Certainly, in the future, the relationship between computer science and bridge damage detection technology will be more intimately connected.

### 6.2.3. Deterioration Model

A deterioration model is a useful tool for service life prediction. They consider traffic loads, climate change, anodic dissolution, and hydrogen embrittlement. However, there are few studies incorporating fully integrated models. Field experiment data measured by sensors is capable of providing accurate deterioration data in fatigue condition assessment. Well-established AI-based models have the potential to avoid some model limitations by overcoming the nascent development stage. Artificial neural network (ANN) based models using multilayer perceptron (MLP) generates missing condition state data to fill the gaps created by irregular inspections.

## 7. Conclusions

This paper has outlined the body of work associated with UAS damage detection technology for life prediction with the aim of providing a current status for researchers. The current development and recent advances with future capabilities are demonstrated through the literature review. Owing to an influx of UAS technologies, improved sensors, and efficient algorithms, the field of damage detection is rapidly developing. All infrastructure, including but not limited to bridges, is filled with unpredictable risks, especially relevant to human safety. UAS cooperated sensors make use of advances in damage detection, permitting inspectors to gather complete damage information and to perform a reliability analysis in the context of the above studies. To achieve applicable systems, fundamental insights and technological breakthroughs are needed in the following areas:

(i). robust autonomy: how to produce verifiable and scrutiny-proof algorithms that lead to desired emerging outcomes with real-time damage detection.

(ii). UAS flight strategy: facilitating high-efficiency damage information extracts and flight path manipulations, depending on UAS technology breakthroughs.

(iii). reliable model: enabling output feedback and evaluation of structures and stability based on field measurement data.

(iv). system integration: insights into the co-design of hardware and software and a combination of bottom-up and top-down strategies may be combined and leveraged for enhanced functionality.

The field still requires many different studies. Although this review mainly focuses on bridges, many of the damage detection technologies can also be applied to other infrastructures. Moving forward, future research will also reveal the role of physical humans with respect to UAS damage detection. They may define flight strategy, offer online corrections, or maintain and support methods for the UAS. The emerging technology covered in these systematic scientific studies of UAS damage detection will undoubtedly unlock new understanding critical to both UAS systems and sensor technology, along with addressing integrated systems, data processing, service life modeling, and properties, simultaneously.

**Author Contributions:** Conceptualization, H.L.; formal analysis, H.L.; writing—original draft, H.L.; supervision, Y.C. and H.Z.; visualization, J.L. and Z.Z.; resources, J.L.; funding acquisition, Y.C.; validation, Y.C. and H.Z.; project administration, H.Z.; writing—review and editing, H.Z. All authors have read and agreed to the published version of the manuscript.

**Funding:** This research was funded by Jilin Province Key R&D Plan Project under Grant: 20200401113GX.

**Data Availability Statement:** Not applicable.

**Conflicts of Interest:** The authors declare no conflict of interest.

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
