# Peer review of "Unmanned Aircraft System Applications in Damage Detection and Service Life Prediction for Bridges: A Review"

_remotesensing, doi:10.3390/rs14174210_

Round 1
Reviewer 1 Report
The authors summarize the present research of bridge damage detection using unmanned aircraft system. The manuscript compared the different sensors and the different processing methods, and indicate their advantages shortcomings. It is well organized and helpful for understanding the field of bridge damage detection.
Several minor comments:
Line 215: there is space within the caption of section 3.1.3.
Line 218: ". image based..." should be ": image based..."
There are several "Error! Reference source not found". Please delete or revise them.
Author Response
Response to Reviewer 1 Comments
Thank you for your review report concerning our manuscript entitled “Unmanned aircraft system applications in damage detection and service-life prediction for bridges: A review” (ID: 1841153). Those comments are all valuable and helpful for revising and improving our paper. The main corrections in the paper and in-depth response to your comments are as following:
Reviewer 1 |
|
Comment 1 |
The authors summarize the present research of bridge damage detection using unmanned aircraft system. The manuscript compared the different sensors and the different processing methods, and indicate their advantages shortcomings. It is well organized and helpful for understanding the field of bridge damage detection. |
Response |
Thanks for your approval to our manuscript. |
Comment 2 |
Several minor comments: Line 215: there is space within the caption of section 3.1.3. |
Response |
The meaning of “3.1.3.” is section 3.1. and 3D reconstruction and the title has revised to ‘’Three-dimensional Reconstruction‘’ to avoid misunderstanding. |
Comment 3 |
Line 218: ". image based..." should be ": image based..." |
Response |
It is truly as you said and we have revised. |
Comment 4 |
There are several "Error! Reference source not found". Please delete or revise them. |
Response |
We have checked reference error and the error was not shown in our pdf version. We have re-edited the error reference in our manuscript and we will seek further help from editors if the errors are still existed. |
We tried our best to improve the manuscript and made some changes in the manuscript. These changes will not influence the content and framework of the paper.
We would like to express our great appreciation to editors and reviewers for the comments on our paper. Looking forward to hearing from you.
Thank you and best regards.
Yours sincerely,
Hongze Li
8.8.2022

Reviewer 2 Report
- Nonetheless, it is a survey paper, but it should categorize the topic well and provide some degrees of innovation and interesting conclusions.
- The punctuation and grammar need to be revised and double-checked. Examples: “concise, but though” ïƒ thorough. “UAV 419 self-controlled navigation is still a not a little challenge in the domain of bridge inspection”
- As a general rule, put one space after a period, comma, colon, semicolon, etc. Use a space before the opening parenthesis or bracket.
- Provide the methodology you choose to be used to survey this subject, either in figure 1 or in a separate figure. This is needed because the topic is wide and comprises several components.
- In section 2.1, provide more information on why these UAV models—rather than many other platforms—are appropriate for this type of inspection. Or to put it in a different way: Do these UAV models essentially have a certain dynamic and control system that makes them qualified for this mission?
- In table 1, please provide the full names of any abbreviations, if there are any. Additionally, the methods' focus is unclear; please, give a clearer explanation. Do the same for the rest of the paper when you need to explain an abbreviation.
- As a general rule, please keep in mind that UAV flight times are still limited (often around 30 minutes), and they may even become shorter if the drone is equipped with additional sensors. The quality of data collection is also influenced by the distance to the target and the UAV's flight speed, which has to be considered.
- Figure 4's description is quite unclear, and the figure’s details are generally unnecessary unless the authors plan to explain it completely.
- Please add some link paragraphs to connect the several parts that are now unconnected from one another. For instance, take a look at the transition from section 3 to section 4.
- Particularly in the second half of the paper where there are many different topics mentioned, the authors lose control of the work.
- Since no numerical comparison is offered to assess the benefits and drawbacks of each approach, it is hard to conclude at the end of the paper.
Author Response
Response to Reviewer 2 Comments
We deeply appreciate the time and great effort you spent in reviewing our manuscript entitled “Unmanned aircraft system applications in damage detection and service-life prediction for bridges: A review” (ID: 1841153). The comments are all important and very helpful for us to revise and improve our manuscript. We have considered these comments carefully and have tried our best to revise our manuscript according to the comments. The comments have been responded in point-by-point manner and are listed as follows:
Reviewer 2 |
|
Comment 1 |
Nonetheless, it is a survey paper, but it should categorize the topic well and provide some degrees of innovation and interesting conclusions. |
Response |
Thanks for your advice and we have tried to improve our manuscript. |
Comment 2 |
The punctuation and grammar need to be revised and double-checked. Examples: “concise, but though” ïƒ thorough. “UAV 419 self-controlled navigation is still a not a little challenge in the domain of bridge inspection” |
Response |
Revised. We are very sorry for our negligence and we have checked through our manuscript. |
Comment 3 |
As a general rule, put one space after a period, comma, colon, semicolon, etc. Use a space before the opening parenthesis or bracket. |
Response |
It is truly as you said there are some inappropriate places in our manuscript and we have revised them. |
Comment 4 |
Provide the methodology you choose to be used to survey this subject, either in figure 1 or in a separate figure. This is needed because the topic is wide and comprises several components. |
Response |
Revised. We provide the flow chart of the methodology in Figure 1. |
Comment 5 |
In section 2.1, provide more information on why these UAV models—rather than many other platforms—are appropriate for this type of inspection. Or to put it in a different way: Do these UAV models essentially have a certain dynamic and control system that makes them qualified for this mission? |
Response |
We are much obliged about your suggestion. Our idea can be conveyed more clearly by adding a separate paragraph covered UAV advantages compared with other platforms. |
Comment 6 |
In table 1, please provide the full names of any abbreviations, if there are any. Additionally, the methods' focus is unclear; please, give a clearer explanation. Do the same for the rest of the paper when you need to explain an abbreviation. |
Response |
Done. We provide the full names in footnote and add a row to explain methods’ focus for all tables. |
Comment 7 |
As a general rule, please keep in mind that UAV flight times are still limited (often around 30 minutes), and they may even become shorter if the drone is equipped with additional sensors. The quality of data collection is also influenced by the distance to the target and the UAV's flight speed, which has to be considered. |
Response |
Thanks for your professional advice and we have expressed your idea to readers in Section 6.1. |
Comment 8 |
Figure 4's description is quite unclear, and the figure’s details are generally unnecessary unless the authors plan to explain it completely. |
Response |
We decide to added the figure’s details for readers to understand basic rules of deep learning in bridge damage detection after a brief discuss. |
Comment 9 |
Please add some link paragraphs to connect the several parts that are now unconnected from one another. For instance, take a look at the transition from section 3 to section 4. |
Response |
Added. We have added some link paragraphs in Section 4. |
Comment 10 |
Particularly in the second half of the paper where there are many different topics mentioned, the authors lose control of the work. |
Response |
We have tried our best to re-edit this part in our revised manuscript. |
Comment 11 |
Since no numerical comparison is offered to assess the benefits and drawbacks of each approach, it is hard to conclude at the end of the paper. |
Response |
We have added some numerical results in Section 3. |
We tried our best to improve the manuscript and made some changes in the manuscript. These changes will not influence the content and framework of the paper.
We would like to express our great appreciation to editors and reviewers for the comments on our paper. Looking forward to hearing from you.
Thank you and best regards.
Yours sincerely,
Hongze Li
8.8.2022

Reviewer 3 Report
The topic of the reviewed article is highly topical. The authors performed a meritorious activity and mapped out a solution to a selected complex interdisciplinary problem, namely the application of new technologies to solve the diagnosis of the technical condition of bridges.This review provides an overview of crucial scientific advances that covered the development of UAS inspection: underlying UAS platforms, peripherals, sensing equipment, data processing approach, and service-life prediction models. This review also includes highlights of the remaining scientific challenges and development trends, including the critical need for self-navigated control, autonomic damage detection, and deterioration model building. The article also includes an interesting expert discussion of the pros and cons of this common technology, along with a research perspective on UAS damage detection technology. The article also contains a rich source of references, which covers several scientific areas, from which experts dealing with this issue must draw. While studying the materials related to the opponent's review of this article, I also got acquainted with the content of an interesting study that was conducted at the University of Twente in the Netherlands, I would allow myself to recommend the authors to include it in the rich list of references of this review article.
Author: Jongerius, Alexander (alexander.jongerius@gmail.com):
Title: The use of unmanned aerial vehicles to inspect bridges for Rijkswaterstaat
Educational institution: University of Twente Faculty of Engineering Technology Civil Engineering
Date: March 25, 2018
I have the following comment on the article. I consider UAV flight path programming to be an important part of UAS preparation for the application of diagnostics of more complex bridge structures. I would consider it appropriate if the authors reserved a separate paragraph for this area and listed the most important articles in the list of references.. I positively assess that the authors managed to fulfill their goal in a relatively small space to provide an overview of important works connected with the development of UAS inspection underlying UAS platforms and etc
From the formal comments, I state that there are several erroneous references to the literature in the pdf format of the article. I am attaching the assessed article with marked errors.

Author Response
Response to Reviewer 3 Comments
Thank you for your review report concerning our manuscript entitled “Unmanned aircraft system applications in damage detection and service-life prediction for bridges: A review” (ID: 1841153). Those comments are all valuable and helpful for revising and improving our paper, as well as the important guiding significance to our research. We have studied comments carefully and have made a lot of correction which we hope meet with approval. The main corrections in the paper and in-depth response to your comments are as following:
Reviewer 3 |
|
Comment 1 |
While studying the materials related to the opponent's review of this article, I also got acquainted with the content of an interesting study that was conducted at the University of Twente in the Netherlands, I would allow myself to recommend the authors to include it in the rich list of references of this review article. Author: Jongerius, Alexander (alexander.jongerius@gmail.com): Title: The use of unmanned aerial vehicles to inspect bridges for Rijkswaterstaat Educational institution: University of Twente Faculty of Engineering Technology Civil Engineering Date: March 25, 2018 https://essay.utwente.nl/74894/1/Jongerius-Alexander.pdf |
Response |
It is really true as you suggest that the paper is well organized and helpful for understanding the field of UAV bridge inspection. We have added it to our references in Line 53. |
Comment 2 |
I consider UAV flight path programming to be an important part of UAS preparation for the application of diagnostics of more complex bridge structures. I would consider it appropriate if the authors reserved a separate paragraph for this area and listed the most important articles in the list of references.. |
Response |
As you said, UAV flight path is another important task when applying this technology. We have added two separate paragraphs in Section 2.1, which covered two path planning methods applied to bright inspection surveys and several common algorithms applying for UAV flight path programming. |
Comment 3 |
From the formal comments, I state that there are several erroneous references to the literature in the pdf format of the article. I am attaching the assessed article with marked errors. |
Response |
Thanks for your help. We have checked reference error and the error was not shown in our pdf version. We have re-edited the error reference in our manuscript and we will seek further help from editors if the errors are still existed. |
We tried our best to improve the manuscript and made some changes in the manuscript. These changes will not influence the content and framework of the paper.
We would like to express our great appreciation to editors and reviewers for the comments on our paper. Looking forward to hearing from you.
Thank you and best regards.
Yours sincerely,
Hongze Li
8.8.2022

Round 2
Reviewer 2 Report
The authors performed a great job of revising the manuscript and taking my comments into account. The paper is generally in good form and appears to be prepared for publication, in my opinion.
I have a couple of minor suggestions.
1- The "regulation" subsection could be expanded to include information on UAVs' capacity to fly close to places of interest. Are there any variations in how UAVs are deployed for this type of inspection between nations, for instance, across the US, Europe, Asia, and Africa. We know that difference exists regarding the maximum altitude, minimum distance from people, etc. In summary, it would be beneficial (for the folks with practical interest in this type of inspection) to include any necessary information/citation comparing them in this survey if there are any.
2- Please make sure you edit and proofread the text before submitting.
Author Response
Response to Reviewer 2 Comments
Thanks again for your review report concerning our manuscript entitled “Unmanned aircraft system applications in damage detection and service-life prediction for bridges: A review” (ID: 1841153). Those comments are all valuable and helpful for revising and improving our paper, as well as the important guiding significance to our manuscript. The main corrections in the paper and in-depth response to your two comments are as following:
Reviewer 2 |
|
Comment 1 |
The "regulation" subsection could be expanded to include information on UAVs' capacity to fly close to places of interest. Are there any variations in how UAVs are deployed for this type of inspection between nations, for instance, across the US, Europe, Asia, and Africa. We know that difference exists regarding the maximum altitude, minimum distance from people, etc. In summary, it would be beneficial (for the folks with practical interest in this type of inspection) to include any necessary information/citation comparing them in this survey if there are any. |
Response |
It is really true as you suggest that UAV regulation is a main consideration. We have expanded by thumbing through technical literatures about this subject. |
Comment 2 |
Please make sure you edit and proofread the text before submitting. |
Response |
We have checked our manuscript again and correct some little mistakes. |
We tried our best to improve the manuscript and made some changes in the manuscript.
We would like to express our great appreciation to editors and reviewers for the comments on our paper. Looking forward to hearing from you.
Thank you and best regards.
Yours sincerely,
Hongze Li
8.11.2022
